# Proteomics-Based Machine Learning Approach as an Alternative to Conventional Biomarkers for Differential Diagnosis of Chronic Kidney Diseases

**DOI:** 10.3390/ijms21134802

**Published:** 2020-07-07

**Authors:** Yury E. Glazyrin, Dmitry V. Veprintsev, Irina A. Ler, Maria L. Rossovskaya, Svetlana A. Varygina, Sofia L. Glizer, Tatiana N. Zamay, Marina M. Petrova, Zoran Minic, Maxim V. Berezovski, Anna S. Kichkailo

**Affiliations:** 1Laboratory for Biomolecular and Medical Technologies, Krasnoyarsk State Medical University Named after Prof. V.F. Voyno-Yasenetsky, 660022 Krasnoyarsk, Russia; tzamay@yandex.ru (T.N.Z.); annazamay@yandex.ru (A.S.K.); 2Laboratory for Digital Controlled Drugs and Theranostics, Federal Research Center “Krasnoyarsk Science Center of the Siberian Branch of the Russian Academy of Science”, 660036 Krasnoyarsk, Russia; d_veprintsev@mail.ru; 3Department of Nephrology, Krasnoyarsk Interdistrict Clinical Hospital of Emergency Medical Care Named after N.S. Karpovich, 660062 Krasnoyarsk, Russia; irina-ler@bk.ru (I.A.L.); mross@mail.ru (M.L.R.); alfasv-ja@list.ru (S.A.V.); sofiaglizer@mail.ru (S.L.G.); 4Faculty of Medicine, Krasnoyarsk State Medical University Named after Prof. V.F. Voyno-Yasenetsky, 660022 Krasnoyarsk, Russia; stk99@yandex.ru; 5Department of Chemistry and Biomolecular Sciences, University of Ottawa, Ottawa, ON K1N6N5, Canada; zminic@uottawa.ca (Z.M.); Maxim.Berezovski@uottawa.ca (M.V.B.)

**Keywords:** chronic kidney disease, machine learning, differential diagnosis, proteomics, mass spectrometry, label-free quantification

## Abstract

Diabetic nephropathy, hypertension, and glomerulonephritis are the most common causes of chronic kidney diseases (CKD). Since CKD of various origins may not become apparent until kidney function is significantly impaired, a differential diagnosis and an appropriate treatment are needed at the very early stages. Conventional biomarkers may not have sufficient separation capabilities, while a full-proteomic approach may be used for these purposes. In the current study, several machine learning algorithms were examined for the differential diagnosis of CKD of three origins. The tested dataset was based on whole proteomic data obtained after the mass spectrometric analysis of plasma and urine samples of 34 CKD patients and the use of label-free quantification approach. The k-nearest-neighbors algorithm showed the possibility of separation of a healthy group from renal patients in general by proteomics data of plasma with high confidence (97.8%). This algorithm has also be proven to be the best of the three tested for distinguishing the groups of patients with diabetic nephropathy and glomerulonephritis according to proteomics data of plasma (96.3% of correct decisions). The group of hypertensive nephropathy could not be reliably separated according to plasma data, whereas analysis of entire proteomics data of urine did not allow differentiating the three diseases. Nevertheless, the group of hypertensive nephropathy was reliably separated from all other renal patients using the k-nearest-neighbors classifier “one against all” with 100% of accuracy by urine proteome data. The tested algorithms show good abilities to differentiate the various groups across proteomic data sets, which may help to avoid invasive intervention for the verification of the glomerulonephritis subtypes, as well as to differentiate hypertensive and diabetic nephropathy in the early stages based not on individual biomarkers, but on the whole proteomic composition of urine and blood.

## 1. Introduction

Chronic kidney disease (CKD) is a supra-nosological concept that unites all patients with signs of kidney damage and/or a decrease in their function [1]. CKD is one of the major health problems with high mortality, because it causes irreversible changes in renal failure. No obvious clinical symptoms appear in early stage disease until severe damage has occurred [2]. Therefore, the need for early diagnosis of CKD is obvious. Diseases leading to CKD can be divided into two groups: (1) processes localized directly in the kidneys and urinary tract (glomerulonephritis, pyelonephritis, etc.), and (2) diseases in which the kidneys are target organs (diabetes, hypertensive disease, systemic diseases, etc.). Diagnosis of the disease causing the damage is paramount in all cases of the CKD presence [3,4,5]. The most common causes of CKD are diabetic nephropathy, hypertension, and glomerulonephritis [6]. Clinical manifestations, serum creatinine (Scr), and renal histopathology are commonly used to diagnose CKD and determine its different stages. The role of Scr is very limited [2]. Although kidney biopsy for histopathology may be an invasive and painful procedure, it is considered as the gold standard for the diagnosis of renal disease [7]. Bleeding and other surgical complications may follow this procedure. To reduce these risks, it could be safer to use alternative techniques.

The study of proteomic composition of urine and other human bio-fluids is very promising for the diagnosis of different kidney pathologies and for understanding the mechanisms of their occurrence. Proteinuria may reflect abnormal plasma protein loss, as a result of: (a) an increase in glomerular permeability for macromolecular proteins (glomerular proteinuria), (b) incomplete tubular reabsorption of low molecular weight proteins (tubular proteinuria), (c) abnormal loss of proteins of renal origin and urinary tract. Thus, the analysis of the urine proteome potentially allows us to speak about the localization of nephron damage, which greatly facilitates the differential diagnosis [8,9]. Research is currently underway, both in the search for specific proteins found in CKD [10,11,12,13,14] and attempts to highlight individual proteins that would become markers of specific diseases that cause CKD [15,16].

Often, information on changes in the expression level of a single protein is not enough to obtain sufficient accuracy and the sensitivity required for a clinical diagnostic system, and it is necessary to apply several indicators simultaneously. Thus, the use of a panel of 28 urinal proteins has shown the ability to differentiate Immunoglobulin-A nephropathy and primary membranous nephropathy with a sensitivity of 77% and specificity of 100% [17]. The sets of differently expressed urinal proteins were used for the differential diagnosis of lupus nephritis, primary membranous nephropathy, diabetic nephropathy, and focal segmental glomerulosclerosis. The sensitivity of differential diagnosis remained at 70% when using a set of 5 proteins, but the accuracy fell below 50% when using a set of less than 20 proteins [18]. It shows that these indicators are still insufficient for the effective differentiation of CKDs.

However, the most versatile and universal approach for differential diagnosis should consider the full quantitative information about a large number of proteins contained in patient’s fluids. Multicomponent proteomics data derived from mass spectrometric analysis of a non-diagnosed patient’s sample can be processed in comparison with similar data sets obtained from people with known diagnoses, to assign a new patient to a particular group. For this purpose the mathematical models of machine learning, which take into account the interactions of a large amount of data in a multidimensional space, can be used. Such an approach may become a new concept of an effective and universal test system for both early diagnosis of CKD and post diagnostic differentiation of renal diseases of different origin.

Recent methods of large data sets processing are often based on the principle of “black box”, where input data are transformed into decision factors without any additional knowledge of internal working. Mathematical instruments, such as machine learning and data analysis, are increasingly being used in medicine [19,20]. Machine learning is a branch of the data science that trains computers to perform tasks by observing patterns in large datasets and using them to derive rules or algorithms that optimize task performance [21]. It is used for computer-aided diagnosis of acute neurological events [22] and retinal disease [23]. These studies were mainly based on general clinical indicators, whereas the application of the wide-scale method of quantitative proteomics based on a comparison of relative expressions of a large number of proteins can show much greater efficiency.

In this paper, we introduce a new approach to the differential diagnosis of CKDs of different origins, such as diabetic nephropathy, chronic glomerulonephritis and hypertensive nephropathy, which is based on large proteomics data sets obtained by mass spectrometry of blood plasma and urine, by means of several models of machine learning. The tested algorithms showed good abilities to differentiate the various groups of the tested renal patients according to the proteomic data.

## 2. Results

Plasma and urine samples were collected from 15 patients with diabetic nephropathy (group D), from 14 patients with glomerulonephritis (group G), from 5 patients with hypertensive nephropathy (group H), and from 14 healthy volunteers (group N). Table 1 summarizes the principal characteristics of the CKD patients.

The plasma samples were depleted from albumin and immunoglobulin, the urine samples were concentrated by filter columns. All protein samples were prepared for mass spectrometry in duplicates and each duplicate was analyzed twice. These technical replicates were marked as A and B. The total set of plasma probes contained 190 samples (group G—56, group D—58, group H—20, and group N—56). The number of urine samples derived from patients was 97 (group G—40, group D—41, and group H—16); urine from healthy people was not taken due to low normal protein concentrations. The samples were not divided into fractions to limit the resulted data set for the convenience of bioinformatic processing. Two sets of quantitative proteomics data from blood and urine samples (Appendix A) were obtained. No specific differences on proteome profiles between different patients groups expressed in the distribution of individual proteins were found by conventional statistical methods. Thus, machine learning algorithms, taking into account the overall contribution of the proteomic composition of the samples, were applied. The data were independently tested in two ways. At first, we distinguished the total CKD patients from the healthy individuals, and then we differentiated three groups of patients from each other without comparison with healthy control.

### 2.1. Separation of CKD Patients from the Healthy Individuals by Plasma Proteome

A total of 246 proteins were identified and quantified by label-free quantification (LFQ) from the plasma probes, and 184 of them were considered as relevant after the quality filtration. Then, the differences in plasma proteome of CKD patients and healthy individuals were estimated by principal component analysis (PCA). After the consistency check and averaging of the replicates, the dataset was reduced to 90 samples (group G—27, group D—28, group H—9, and group N—26) and label-free quantification (LFQ) values of 184 proteins were converted to the 17 principal components providing the cumulative variance of 70%.

The KNeighbors machine learning model (kNN) with the Euclidean distance was used for the separation of the total set of CKD patients (groups G, D, H) from the healthy individuals (group N). The best number of neighbors was found as 3. The mean proportion of correct classifier responses was 97.8%. Thus, the total CKD patients were differentiated from the healthy individuals using proteomics data of plasma with high confidence.

### 2.2. Differentiation of the Three Groups of Renal Patients by Plasma Proteome

The possibility of differentiation of various types of CKD with similar symptoms by proteomics data was tested. At this step, the control samples obtained from healthy people (group N) were discarded and 134 CKD patient’s samples were taken; 175 proteins in 131 samples were left after the quality control check.

The PCA analysis showed that 14 principal components were necessary for 70% of cumulative variance, and 64 averaged results were obtained after the consistency check. Three models of machine learning were tested: KNeighbors, logistic regression, and support vector machine (SVM). The optimal hyper parameters were found: number of neighbors (*n*_neighbors = 1) for KNeighbors, constant of regularization (C = 1) for logistic regression, kernel (kernel = ‘rbf’) and gamma (gamma = 0.5) for SVM. The mean proportions of correct classifier responses were 87.5%, 84.3%, 82.8%, respectively. Based on this cross-validation quality assessment, the KNeighbors classifier was chosen as the best for further calculations. A number of errors of the nearest-neighbors algorithm were found for each class for the entire set. Class D had 2 errors from 28 (7.1%), class G had 2 errors from 27 (7.4%), and class H had 4 errors from 9 (44.4%). Thus, this algorithm works well for differential separation of group D from group G, but does not allow distinguishing class H.

Finally, the class H samples were excluded from the analysis and 55 samples were left in the set. The nearest-neighbor algorithm gave only two errors for this set (the proportions of correct decisions were 96.3% for class G and 96.4% for class D). Thus, this algorithm showed good results for the use in a medical test system that can separate glomerulonephritis from diabetic nephropathy based on plasma proteome.

### 2.3. Differential Diagnosis Based on Urine Proteome and Separation of Group H from the Total Patients

A total of 409 proteins were identified and quantified by LFQ, and 241 proteins in 96 samples left after the quality control.

We found 13 principal components to be optimal after the PCA. The consistency check and averaging resulted in 47 samples (group G—19, group D—20, and group H—8). Four machine learning models were tested. A decision tree was added to the three algorithms mentioned above. None of the tested algorithms with optimized hyper parameters showed high proportion of correct decisions. It was lower than 80% overall, thus showing no evident possibility to differentiate the three tested groups on the basis of urine proteome data, and allowing us to conclude that the urine proteome in comparison with plasma proteome has much less differences in the tested groups of patients.

The capabilities of “one against all” models for the separation of classes were tested. It was found that the classifier 1-nn (nearest-neighbor algorithm) gives an accuracy equal to 100% for class H by the entire set of urine proteomics data. It correctly defined all the samples of class H and had no false definitions. Thus, this model showed good capability to separate patients of group H (hypertensive nephropathy) from the remaining groups of renal patients.

## 3. Discussion

Focused on efficient processing of proteomics data, this work had two goals: 1) development of the concept of a non-invasive early-stage test system for general kidney malfunctions, and 2) the differentiation by origin of previously diagnosed renal diseases with similar symptoms. For these, using our data set, we first tried to distinguish the total group of CKD patients from the healthy group, and then to differentiate the three groups of patients from each other.

At the first step, it was important to find out whether the common differences in plasma proteome appear in patients with renal disease and in healthy people. This task was not particularly difficult. It turned out that KNeighbors machine learning model (kNN) is able to differentiate CKD patients from healthy individuals with high confidence (97.8% of correct classifier responses).

Glomerular filtration rate (GFR) remains for today the most operating marker of kidney malfunction, which is usually estimated taking into account endogenous filtration markers like serum creatinine and cystatin C [24], but the accuracy of their measuring is still under consideration [25]. It shows that new proteomic biomarkers may facilitate more accurate and earlier detection of renal pathologies [26]. Machine learning has been successfully applied to proteomics data for classification of samples and identification of biomarkers, and can be used across a wide range of diseases [27]. Thus, the high accuracy in separating a group of renal patients from healthy ones that we demonstrated by processing of plasma proteome data using machine learning proved to be effective for the introduction of this approach into clinical diagnosis.

The second goal of our research was to examine the ability of complex differentiation of various CKDs by plasma proteome data. It is difficult to predict in advance the most efficient processing algorithm for analyzing multidimensional data, thus we tested three models. The KNeighbors classifier (Model 1) was chosen as the most effective after comparing with the logistic regression and support vector machine (SVM), because of the best proportion of correct classifier responses (87.5%). After the less represented group of hypertensive patients was excluded, the proportion of correct decisions increased above 96%, thus showing a high ability to separate diabetic patients with indirect kidney damage from patients with autoimmune-caused internal kidney degradation (glomerulonephritis).

Diseases of different origins, which are not only related to kidneys but expressed in symptoms of kidney degradation, may appear in plasma proteomic composition. According to our results, diabetic nephropathy has a specific proteomic signature in blood which is independent of renal degradation. The distinct changes in the expression of individual proteins, such as monocyte chemoattractant protein-1 (MCP-1) and transforming growth factor-β1 (TGF-β1) [28], or even in panels of several proteins [29] noted previously as predicting the rate of renal function decline in diabetes, may also be accompanied by other more complex changes in the plasma proteome. On the other hand, the hypertonic signature of renal degradation with the damage of the tubules and the interstitium was not expressed in specific changes in the blood proteome, according to our data. Therefore, the proposed approach can be recommended for further development as a medical test system based on plasma proteome that can separate glomerulonephritis from other CKD patients, e.g., diabetics.

Urine sampling is even easier compared to venous blood collection and the sample preparation procedure eliminates the extra step of plasma depletion from major proteins. We tried to use both bio-fluids to obtain the proteomics data and compared them to find the most suitable way for differential diagnosis of the three types of kidney diseases. Urine samples obtained from healthy people were not used in this study, due to much lower protein concentrations compared with renal patients.

As well as for the plasma proteome, we tested kNN, SVM, and logistic regression as means for distinguishing the urine proteome datasets. In addition, the decision tree algorithm has been added to the comparison. However, as a result, none of the tested models gave the proportion of correct decisions above the level of 80%, thus showing no clear ability to differentiate simultaneously the three tested groups based on urine proteome data.

The additional “one against all” model showed the best results in the separation of only one tested group of patients. The classifier 1-nn (nearest-neighbor algorithm, Model 2) gives an accuracy of 100% for class H, showing no false definitions across the entire data set. Processing of urine proteome by machine learning showed the ability of distinguishing hypertensive nephropathy from other renal diseases. Since, here, we used a sample set that is not very representative (only 5 hypertensive patients), this approach should be recommended for further testing on larger groups.

Diabetic nephropathy and proliferative forms of glomerulonephritis have a similar histological picture of diffuse glomerulosclerosis, tubulointerstitial fibrosis, and atrophy, and also variable degrees of hyaline arteriolosclerosis and arterial sclerosis. In hypertensive nephropathy, primary pathological changes in afferent arterioles lead to ischemic damage to the glomerular apparatus [30]. Thus, the degradation of various kidney structures can result in a difference in the transmitted proteins that enter the urine of hypertensive patients. Proteinuria of the same origin, associated with glomerular defects probably did not have specific features depending on the genesis of renal degradation, and therefore diabetes and glomerulonephritis did not appear in urinal protein variations according to our study.

To test the differences in pathological renal filtration in the context of protein size, we compared molecular masses of the proteins found in the urine of patients from the three CKD groups. No distinct specificity for a particular disease was found in our case, as shown in Figure 1. Thus, the differences in the total proteome, which can be traced by machine learning, do not concern the molecular masses and sizes of proteins and are manifested in other complex features.

In this study, we conclude that the urine proteome, compared with the plasma proteome, has much less differences in our tested groups. As a result, urine is not quite suited for universal differential diagnosis in this analytical way, but this approach may remain useful in some cases, for example, to isolate patients with hypertensive nephropathy.

In addition, it is possible to use a two-stage approach to the differential diagnosis of CKD (Figure 2), including a combination of primary proteomic analysis of urine to cut off the hypertensive origin of the disease (using Model 2), followed by proteomic analysis of blood plasma to separate diabetic nephropathy and glomerulonephritis (using Model 1).

While the application of the presented strategy can be limited for the hypertensive nephropathy due to the fact that its diagnosis is mostly clinical and proteinuria is absent in most of the cases, for correctly diagnosed diabetic nephropathy, a specific therapy aimed at correcting sugar levels can be more actively applied. Treatment strategies for glomerulonephritis include immunosuppression with glucocorticosteroids and cytostatics, associated with the risk of serious infectious complications. The similar therapy for patients with diabetes can be dangerous, since they have a higher risk of purulent complications. Generally, differential diagnosis between glomerulonephritis and diabetic nephropathy should be carried out on the basis of kidney biopsy. However, several complicating points exist: (1) kidney biopsy is not always unambiguous, (2) this is an invasive method, (3) kidney biopsy should be performed in a specialized center, (4) to assess the dynamics of the process, a second study is required, which in the case of kidney biopsy is associated with repeated invasive intervention, and (5) the patient may refuse an invasive procedure. In this work, we studied the phenotypes expressed in the proteomes that distinguish the presented groups. All associated pathologies, including diabetes, should be undoubtedly reflected in the phenotypic proteomes, which we examined by choosing groups in this way (with and without diabetes) and dividing them using a machine learning approach.

## 4. Materials and Methods

### 4.1. Patients

A total of 34 CKD patients participated in the study (Table 1). Diagnosis of diabetic nephropathy patients was made clinically by the presence of diabetes; most patients had hypertension. Hypertensive nephropathy was diagnosed clinically by the presence of hypertension without presence of diabetes; no biopsy was performed in these groups. A group of glomerulonephritis patients were diagnosed by biopsy. All participants signed an informed consent. The study was approved by the Ethical Committee of the Krasnoyarsk State Medical University named after Professor V.F. Voyno-Yasenetsky (ethical code 88/2019 of 27 February 2019).

### 4.2. Sample Preparation

The plasma samples were depleted from albumin and immunoglobulin using ProteoPrep depletion columns (Sigma-Aldrich, St. Louis, MO, USA) according to the manufacturer’s protocol. Proteins from the urine samples were concentrated with Amicon Ultra-4 3 kDa centrifugal filter columns (Merck Millipore, County Cork, Ireland). The protein concentrations were determined by UV-1280 spectrophotometer (Shimadzu, Kyoto, Japan) and samples with 4 μg of protein were taken for mass spectrometry. The proteins were reduced by dithiothreitol, alkylated by iodoacetamide, and digested by trypsin according to the manufacturer’s protocols (Thermo Scientific, Waltham, MA, USA). The samples were desalted by 10 μL C18 pipette tips from the same manufacturer in accordance with its protocol and dried before analysis.

### 4.3. Liquid Chromatography and Mass Spectrometry

The samples were dissolved in phase “A” (0.1% of formic acid) and 2 μg of each sample were injected into the Dionex UltiMate 3000 RSLC nano liquid chromatographer (Thermo Scientific, USA) with Acclaim RSLC PepMap C18 separation column (15 cm length, 75 μm inner diameter, 2 μm particles). The solvent gradient was increased from 0% to 40% of phase “B” (0.1% of formic acid in 80% acetonitrile) for 90 min, maintaining a constant flow rate of 200 nL/min. The Orbitrap Fusion mass spectrometer (Thermo Scientific, USA) was operated in data-dependent mode with scans of parent and fragment ions changing in cycle of 4 s. Full scans were made at a resolution of 60,000 by the Orbitrap mass detector, and fragments generated by high energy collision dissociation (HCD) were registered by ion trap at normal rate.

### 4.4. Protein Search and Label-Free Quantification

The raw data files were processed by MaxQuant 1.6 software (Max Planck Institute for Biochemistry, Martinsried, Germany) [31]. Label-free quantification (LFQ) parameter was enabled. Unique and razor peptides were chosen for protein quantification. Carbamidomethylation of cysteine was set as a fixed modification, and oxidation of methionine and N-term acetylation were set as variable modifications. Protein search was performed against an actual SwissProt protein database with the human taxonomy restriction. The Orbitrap instrument was chosen and mass tolerance of 20 ppm was set for the first search and 4.5 ppm was set for the main search. Protein false discovery rate (FDR) of 0.01 was set. The obtained values of LFQ intensities (Appendix A) were considered as relative indicators of protein expressions over the sample groups.

### 4.5. Data Analysis

All LFQ data analysis was performed using Anaconda Python 3 with Pandas and Scikit-learn libraries. Two steps of data filtration were made. Non-widespread proteins and low quality samples were excluded from the further analysis. Proteins detected in less than 5 samples and samples in which less than 50 proteins were found were not taken into account. Then, the LFQ normalization procedure was performed as follows: protein LFQ level for each sample was divided to the maximum of LFQ level of the certain protein among all the samples, giving the quantitative values to the range from 0 to 1. Then principal component analysis (PCA) was applied to reduce the data dimensions. We switched to a lower number of features expressed in projections of multiple protein LFQ values to the vectors of principal components. The next step was performed to check the consistency of the samples using data from the pairs of technical replicates. A closest Euclidean neighbor in the space of the principal components was found for each sample. The replicated samples were considered as consistent if a closest neighbor for replicate A was the corresponding replicate B and vice versa. The coordinates in the space of principal components for these replicates were averaged and they were considered as one sample. Samples that were not verified by this method were excluded.

Machine learning algorithms KNeighbors (kNN), logistic regression, support vector machine (SVM) and decision tree were used at the final step of the data analysis. In order to find optimal model hyper parameters, a grid search was used. As the number of experimental samples was not large enough, we did not use the separation of the dataset into training and test sets. We could not select a test set for the final quality verification of the models, since in this case the evaluation of the quality of the trained model would be very unstable, due to the randomness of the division of training and test sets, when marginal measurements may fall into the test set. Instead, we used the leave-one-out cross-validation on the whole dataset to control the model quality.

Accordingly, we excluded each sample from the entire set, trained the model on the remaining set, and then checked it on this extracted sample.

Then, we used mean accuracy metrics, i.e., the average proportion of correct answers to estimate the quality. This metrics seems to be adequate, because the class imbalance is insignificant (the ratio of class sizes does not exceed 3.2).

These procedures allowed us to use all available data as efficiently as possible and to avoid randomness in choosing a test sample.

## 5. Conclusions

The machine learning algorithms show good abilities to differentiate various groups across large data sets. Full proteomics data obtained by mass spectrometry can be very useful for this approach to medical diagnosis. These data contain general information about changes in normal body processes in those cases when common diagnostic approaches using single biomarkers or even panels of biomarkers do not work well enough. With the testing machine learning models on proteomics data obtained from the plasma and urine of patients of three types of CKD, the best results were obtained using the nearest-neighbor algorithm. In this case, according to the proteomics data of plasma, the two groups of patients with diabetic nephropathy and glomerulonephritis are well separated from the group of healthy people. On the other hand, a less presented group of patients with hypertensive nephropathy is better isolated from groups of patients of the two other CKD types by the “one against all” method based on the urine proteome data set.

The further development of the approach presented here may help to avoid invasive intervention and contraindicated intervention in some cases for the verification of the glomerulonephritis subtypes, which is currently performed only by kidney biopsy and microscopic morphological confirmation. The diagnosis of hypertensive and diabetic nephropathy at an early stage also remains relevant and the capabilities of machine learning methods based on proteomics data may be useful for this purpose.

## Figures and Tables

**Figure 1 ijms-21-04802-f001:**
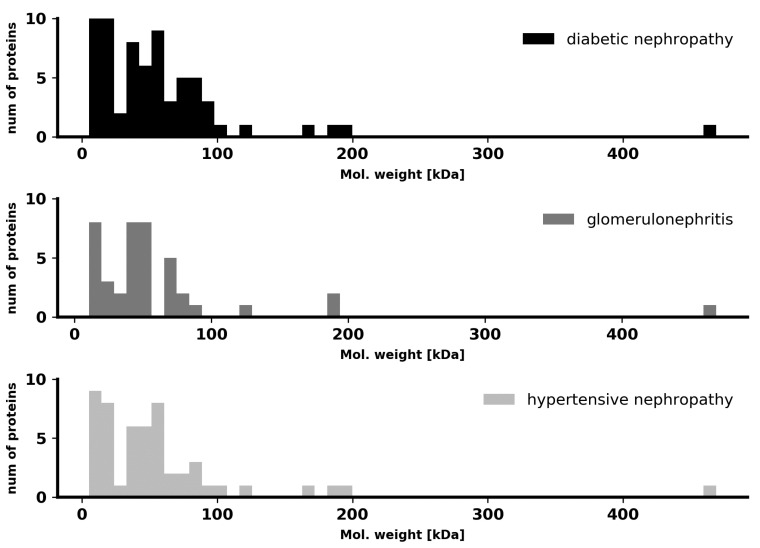
Distribution of urine proteins by molecular weight in the three studied groups of patients.

**Figure 2 ijms-21-04802-f002:**
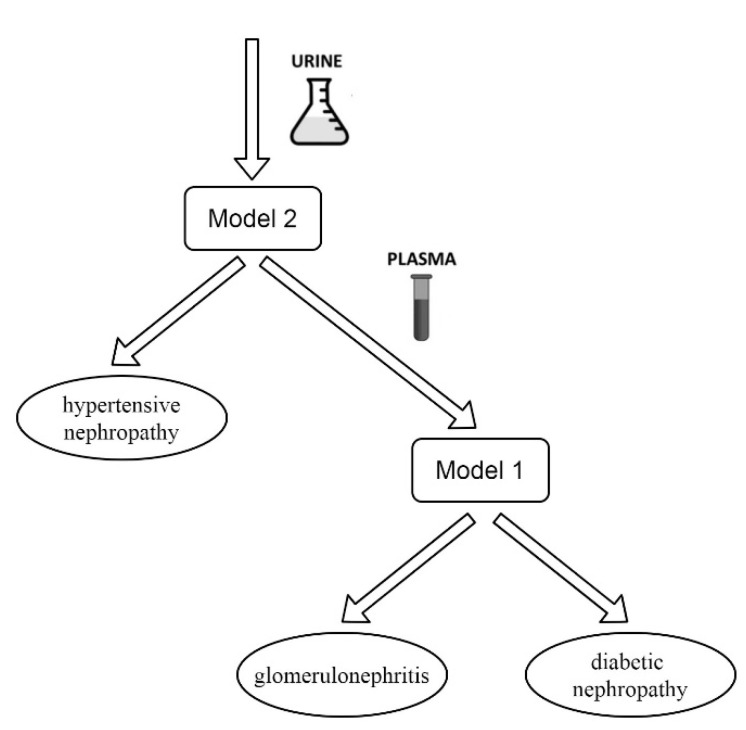
Scheme of two-stage differential diagnosis of chronic kidney disease using proteomic data of urine and plasma.

**Table 1 ijms-21-04802-t001:** Principal characteristics of the chronic kidney diseases (CKD) patients.

Variable	Group D	Group G	Group H
Sex, *n* (%)			
Male	6 (40)	8 (57)	2 (40)
Female	9 (60)	6 (43)	3 (60)
Age, mean ± SD (years)	62.11 ± 17.91	48.46 ± 16.83	60.6 ± 14.15
Stage of renal disease, *n* (%)			
Stage 1	6 (40)	7 (50)	0 (0)
Stage 2 + 3	7 (46.7)	2 (14.3)	0 (0)
Stage 4 + 5	2 (13.3)	5 (35.7)	5 (100)
Level of proteinuria			
Below 30 mg/l, *n* (%)	0 (0)	0 (0)	0 (0)
30 - 300 mg/l, *n* (%)	5 (33.4)	5 (35.7)	1 (20)
Above 300 mg/l, *n* (%)	10 (66.6)	9 (64.3)	4 (80)
Presence of hypertension, *n* (%)	8 (53)	10 (71.4)	5 (100)
eGFR, mean ± SD (mL/min)	39.56 ± 15.27	68.38 ± 48.36	14.8 ± 7.19
Biopsy tested patients, *n* (%)	0 (0)	2 (14.3)	0 (0)
Type of glomerulonephritis			
Chronic glomerulonephritis, *n* (%)		10 (71.4)	
Nephrotic syndrome, *n* (%)		4 (28.6)	

Abbreviations: SD, standard deviation; eGFR, estimated glomerular filtration rate.

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
