# Peer review of "Proteomics-Based Machine Learning Approach as an Alternative to Conventional Biomarkers for Differential Diagnosis of Chronic Kidney Diseases"

_ijms, 2020, doi:10.3390/ijms21134802_

Round 1
Reviewer 1 Report
The authors describe a Proteomics-Based Machine Learning Approach as an Alternative to Conventional Biomarkers for Differential Diagnosis of Chronic Kidney Diseases. It is an interesting manuscript, which could be improved after revision.
Major comments:
Abstract: provides a good summary of the article
Introduction: Generally good, but with respect to the opinions of the
authors regarding biopsy 55-58 , this is not supported by the majority of clinical data
The introduction is clear, useful in setting the scene and understandable but there are few, very minor, errors in sentence construction
Materials and methods section:
Please provide more data on patients groups
- Age, stage of renal disease, vintage of CKD (in advanced CKD, renal dysfunction and reduced clearance may affect proteomics results)
- Proteinuria or not -level of proteinuria
- In different glomerulopathies, on active disease stage, selective proteinuria may be noticed, how this may influence blood/ urine proteomics?
- Patients with glomerulopathies/diabetes were not also hypertensive?
- Hypertensive group (cause of hypertension/ controlled or uncontrolled hypertension)
- Urine of controls should be also analyzed
Results section: it would be interesting to add more detailed data on differences on proteome profile between different patients groups
Conclusions: Whilst I understand the desire to try to derive meaning, the authors would be better avoiding not particularly useful, conclusions regarding renal biopsy. Renal biopsy apart from diagnosis provides also important information regarding the severity of renal involvement. Therefore I believe that renal biopsy cannot avoided in many cases.
Minor comments:
The manuscript will require only minor copy editing during production to formalise the English.
Reviewer 2 Report
Reviewing:
The authors present an interesting solution to a relevant question in clinical practice. In some cases, the distinction between diabetic/vascular nephropathy and primary glomerular disease may require a kidney biopsy. A noninvasive tool could be helpful to avoid biopsy in some cases.
Minor remarks:
-Line 104, 240 the term of hypertonic should be replaced by hypertensive.
-The authors should mitigate the sentence concerning the complications of kidney biopsy
Major remarks:
How the diagnosis of diabetic nephropathy, hypertensive nephropathy, or glomerulonephritis has been made? Has a renal biopsy been performed for each patient? What was the estimated glomerular filtration rate of these patients? In the G group what type of glomerulonephritis were included?
A Table summarizing the principal characteristics of each group should be added for more clarity
The diagnosis hypertensive nephropathy (nephroangiosclerosis) is clinical and proteinuria is absent in most of the case. The interest of this kind of tools seems to be limited in this population.
The distinction between primary glomerulonephritis and diabetic nephropathy is more relevant, as is based on histological findings in some cases. However, the majority of the patients presenting diabetic nephropathy (notably type 1) don’t need a kidney biopsy.
The major limitation of this study is that diabetic patients probably present particular proteomic phenotypes distinguish them from other population of patients. What is the impact of diabetes on the distinction between group D and G? A study comparing 2 groups of diabetic patients, one presenting diabetic nephropathy and a second presenting primary glomerulonephritis, would be probably more informative. This limit should be discussed extensively.
Round 2
Reviewer 1 Report
The major and minor points of my review have been sufficiently responded
I have no further comments
Reviewer 2 Report
The major limitation is now discussed. I have no further comment.